# Nephronectin-integrin α8 signaling is required for proper migration of periocular neural crest cells during chick corneal development

**Justin Ma, Lian Bi, James Spurlin, Peter Lwigale\***

Department of Biosciences, Rice University, Houston, United States

**Abstract** During development, cells aggregate at tissue boundaries to form normal tissue architecture of organs. However, how cells are segregated into tissue precursors remains largely unknown. Cornea development is a perfect example of this process whereby neural crest cells aggregate in the periocular region prior to their migration and differentiation into corneal cells. Our recent RNA-seq analysis identified upregulation of nephronectin (Npnt) transcripts during early stages of corneal development where its function has not been investigated. We found that Npnt mRNA and protein are expressed by various ocular tissues, including the migratory periocular neural crest (pNC), which also express the integrin alpha 8 (Itgα8) receptor. Knockdown of either *Npnt* or *Itgα8* attenuated cornea development, whereas overexpression of *Npnt* resulted in cornea thickening. Moreover, overexpression of Npnt variants lacking RGD-binding sites did not affect corneal thickness. Neither the knockdown nor augmentation of Npnt caused significant changes in cell proliferation, suggesting that Npnt directs pNC migration into the cornea. In vitro analyses showed that Npnt promotes pNC migration from explanted periocular mesenchyme, which requires Itgα8, focal adhesion kinase, and Rho kinase. Combined, these data suggest that Npnt augments cell migration into the presumptive cornea extracellular matrix by functioning as a substrate for Itgα8-positive pNC cells.

**\*For correspondence:**
lwigale@rice.edu

**Competing interest:** The authors declare that no competing interests exist.

## Editor's evaluation

This work investigates the role of extracellular matrix (ECM) component nephronectin (Npnt) and integrin a8 (Itga8) in the migration of periocular mesenchymal cells during vertebrate corneal development. They find that knockdown or overrexpression of Npnt and Itga8 leads to changes in corneal thickness, and their finding suggests that Npnt augments cell migration into the presumptive cornea ECM by functioning as a substrate for Itgα8-positive periocular neural crest.

## Introduction

In vertebrates, the cornea comprises three cellular layers: epithelium, stroma, and endothelium. The corneal epithelium is derived from the ectoderm, whereas the stromal keratocytes and corneal endothelium are derived from cranial neural crest cells (*Lwigale et al., 2005*). In birds and humans, neural crest migration from the periocular region into the presumptive cornea occurs in two waves (*Feneck et al., 2020*; *Hay, 1980*). The first wave forms a monolayer of the nascent corneal endothelium, which is accompanied by a second wave of mesenchymal migration into the acellular matrix between endothelium and ectoderm to form the cornea stroma (*Hay and Revel, 1969*; *Noden, 1978*; *Lwigale et al., 2005*; *Creuzet et al., 2005*; *Feneck et al., 2020*). Several mechanisms pertaining to the role of

growth factors and guidance cues have been identified (*Beebe and Coats, 2000*; *Saika et al., 2001*; *Lwigale and Bronner-Fraser, 2009*; *Choi et al., 2014*), but the functional role of extracellular matrix (ECM)/integrin signaling in establishing the cornea remains unclear.

Among the initial effects of the signaling that takes place between the nascent ocular tissues is the synthesis of the corneal ECM by the corneal epithelium in response to inductive signals from the lens vesicle to form the primary stroma (*Hay and Revel, 1969*; *Hendrix et al., 1982*; *Fitch et al., 1988*). In birds and humans, the primary stroma serves an important role as a scaffold for periocular neural crest (pNC) migration throughout the process of corneal development (*Hay and Revel, 1969*; *Bard and Hay, 1975*; *Quantock and Young, 2008*). It consists of multiple ECM proteins, including hyaluronan (*Toole and Trelstad, 1971*), collagen type I and II (*Hayashi et al., 1988*; *Fitch et al., 1988*), laminin (*Doane et al., 1996*), and fibronectin (Fn) (*Fitch et al., 1991*). Following the second wave of pNC migration into the developing cornea, the primary stroma is concomitantly replaced by the secondary stroma that is synthesized by differentiating stromal keratocytes. The secondary stroma comprises the bulk of the adult cornea and consists of collagens and proteoglycans (*Hayashi et al., 1988*; *Quantock and Young, 2008*) that are arranged in patterns that result in transparency (*Chen et al., 2015*). Over the past decades, the roles of collagens and proteoglycans have been the focus of several investigations due to their indispensable functions in corneal transparency. However, most of these studies were conducted in rodents, in which early corneal development does not involve the primary stroma (*Pei and Rhodin, 1970*; *Haustein, 1983*; *Feneck et al., 2019*). Although the ECM plays a critical role during organogenesis by providing cell adhesion substrate, sequestering signaling molecules, providing structural support and mechanical cues (*Hynes, 2014*), the function of the primary stroma during early corneal development remains to be elucidated.

Our recent RNA-seq analysis of pNC differentiation into corneal cells identified novel expression and upregulation of nephronectin (Npnt) transcripts during corneal development. Npnt was discovered as an ECM ligand for integrin α8β1 (α8β1) during mouse kidney development, consisting of 70–90 kDa proteins (*Brandenberger et al., 2001*) generated by alternate splicing (561–609 amino acids). At the same time, it was also identified as preosteoblast epidermal growth factor (EGF)-like repeat protein with meprin A5 protein and receptor protein-tyrosine phosphatase μ domain (POEM) (*Morimura et al., 2001*). Npnt consists of five EGF-like domains in the N-terminal, a central region containing an Arg-Gly-Asp (RGD) sequence, and a meprin-A5 protein-receptor protein tyrosine phosphatase μ (MAM) domain in the C-terminal. The EGF-like and RGD domains have been functionally characterized and shown to play critical roles in development and tissue homeostasis. The RGD domain signals through α8β1 receptor during epithelial–mesenchymal interactions involved in kidney development and maintenance (*Miner, 2001*; *Linton et al., 2007*; *Müller et al., 1997*; *Sato et al., 2009*; *Cheng et al., 2008*; *Inagi et al., 2017*; *Müller-Deile et al., 2017*; *Zimmerman et al., 2018*). These observations in mice were recently confirmed by the identification of recessive mutation in the Npnt gene that caused bilateral kidney agenesis in human fetuses (*Dai et al., 2021*). The EGF-like domains of Npnt are associated with inducing differentiation and proliferation in osteoblasts and dental stem cells through the activation of mitogen-activated protein kinase (MAPK) pathways (*Fang et al., 2010*; *Kahai et al., 2010*; *Arai et al., 2017*) and to induce vascular endothelial cell migration via phosphorylation of extracellular signal-regulated kinases (ERK) and p38MAPK (*Kuek et al., 2016*). Recent studies have also implicated Npnt to play a role in heart development (*Patra et al., 2011*), attachment of the arrector pili muscle to hair follicles (*Fujiwara et al., 2011*), and during forelimb formation in amphibians (*Abu-Daya et al., 2011*). Other studies have also demonstrated that Npnt plays a potential role in diseases such as chronic obstructive pulmonary disease (*Saferali et al., 2020*), diabetic glomerulosclerosis (*Nakatani et al., 2012*), Fraser syndrome (*Kiyozumi et al., 2012*), and in various cancers (*Magnussen et al., 2021*). Despite the pleiotropic functions of Npnt in development and disease, its expression and function have not yet been described in the cornea.

Inspired by our observation that Npnt transcripts were upregulated during corneal development, we sought to establish its role during pNC migration into the developing avian cornea. We first characterized the spatiotemporal expression of Npnt mRNA and protein at stages that correspond with pNC migration. Next, we used replication-competent ASLV long terminal repeat with a splice acceptor (RCAS)-mediated gene knockdown to investigate the role of *Npnt*. We identified Itgα8 as a potential receptor for Npnt during cornea development and confirmed its role in pNC using an in vitro migration assay coupled with inhibitors for α8β1, focal adhesion kinase (FAK), Rho signaling pathway,

and in vivo using gene knockdown. Lastly, we performed misexpression studies using either the full-length, RGD mutant, and truncated versions of Npnt to further confirm its role during pNC migration and that it functions through the RGD domain. Together, our findings show that Npnt secreted into the ECM of the nascent cornea provides a substrate that promotes migration of Itgα8-positive pNC during development.

## Results

### Npnt mRNA and protein are expressed during early ocular development

Our previous RNA-seq analysis of gene expression during pNC differentiation into corneal cells identified upregulation of *Npnt* transcripts during chick corneal development (*Bi and Lwigale, 2019*). To define the expression of Npnt mRNA and protein distribution during this process, we performed section in situ hybridization and immunohistology, respectively. Consistent with our RNA-seq data (*Bi and Lwigale, 2019*), we found that *Npnt* is undetectable in the pNC by embryonic day (E) 4 and in the presumptive corneal endothelium at E5, but it is primarily expressed in the retinal pigment epithelium (RPE) and presumptive lens fiber cells at these time points (*Figure 1A and B*). However, by E6, we found that in addition to the RPE and lens, *Npnt* is vividly expressed by the second wave of migratory pNC that eventually differentiate into the stromal keratocytes (*Figure 1C*, arrowhead). Expression of *Npnt* persisted in the corneal stroma through E12 (*Figure 1—figure supplement 1A–C*), but it is undetectable at E15 (data not shown).

Because Npnt is a secreted protein that plays a role in cell migration (*Kuek et al., 2016*; *Magnussen et al., 2021*; *Yamada and Kamijo, 2016*), we examined its localization at similar time points during cornea development. At E4, Npnt is localized in the optic cup, lens epithelium, surrounding the pNC cells adjacent to the nascent cornea (*Figure 1D–F*, arrow), and in the primary stroma of the cornea (*Figure 1D–F*, asterisk). At E5, strong Npnt staining is observed in the primary stroma (*Figure 1G–I*, asterisk). By E6, vivid Npnt staining was localized in the anterior region of the cornea and diffusely throughout the primary stroma (*Figure 1J–L*). At this time, the migratory pNC also stained positive for Npnt (*Figure 1L*, arrowheads). Strong expression of Npnt persists in the basement membrane of the corneal epithelium through (*Figure 1—figure supplement 1C and D*) and is also faintly detectable in the corneal epithelial cells by E12 (*Figure 1—figure supplement 1E*). Interestingly, the protein expression did not correlate with the strong mRNA expression in the stroma during later stages of cornea development. This mismatch could be due to post-transcriptional regulation that prevents protein expression. It is also possible that post-translational modification by enzymes such as matrix metalloproteinases (MMPs), which are temporally and spatially regulated in the corneal ECM during development (*Huh et al., 2007*), could lead to protein degradation. A previous study showed that Npnt can be modified by MMP cleavage (*Toraskar et al., 2019*). Given that the changes in protein localization occur after the second wave of migration, we can conclude from our results that the expression of Npnt coincides with pNC ingression into the cornea, implicating a potential role during development.

### Knockdown of *Npnt* disrupts corneal thickness during early development

To investigate the function of Npnt during early corneal development, we generated several RCAS-GFP-Npnt-shRNA (*Npnt^{kd}*) constructs in chick DF-1 cells (*Figure 2—figure supplement 1*). The construct with the highest knockdown of *Npnt* was used for the experiments. Constructs containing only GFP (RCAS-GFP) or scrambled short hairpin (sh)RNA (RCAS-GFP-Scr-shRNA) were used as control. The viral constructs were injected to cover the entire cranial region of HH7-8 (*Hamburger and Hamilton, 1951*) chick embryos (*Figure 2A*). Using this approach, the viral constructs affect all the ocular regions where *Npnt* is expressed, including the neural progenitors of the optic cup and neural crest cells, and the ectodermal progenitors of the lens (*Figure 1A–C*). Embryos were reincubated and collected at E7, then the anterior eyes were screened for GFP as an indicator for the extent of viral infection. Only the eyes that showed robust GFP expression were evaluated for knockdown of *Npnt* expression (*Figure 2A*). Histological analysis of E7 corneas based on hematoxylin and eosin (H&E) staining revealed that knockdown of *Npnt* (N = 4) resulted in reduction of corneal thickness

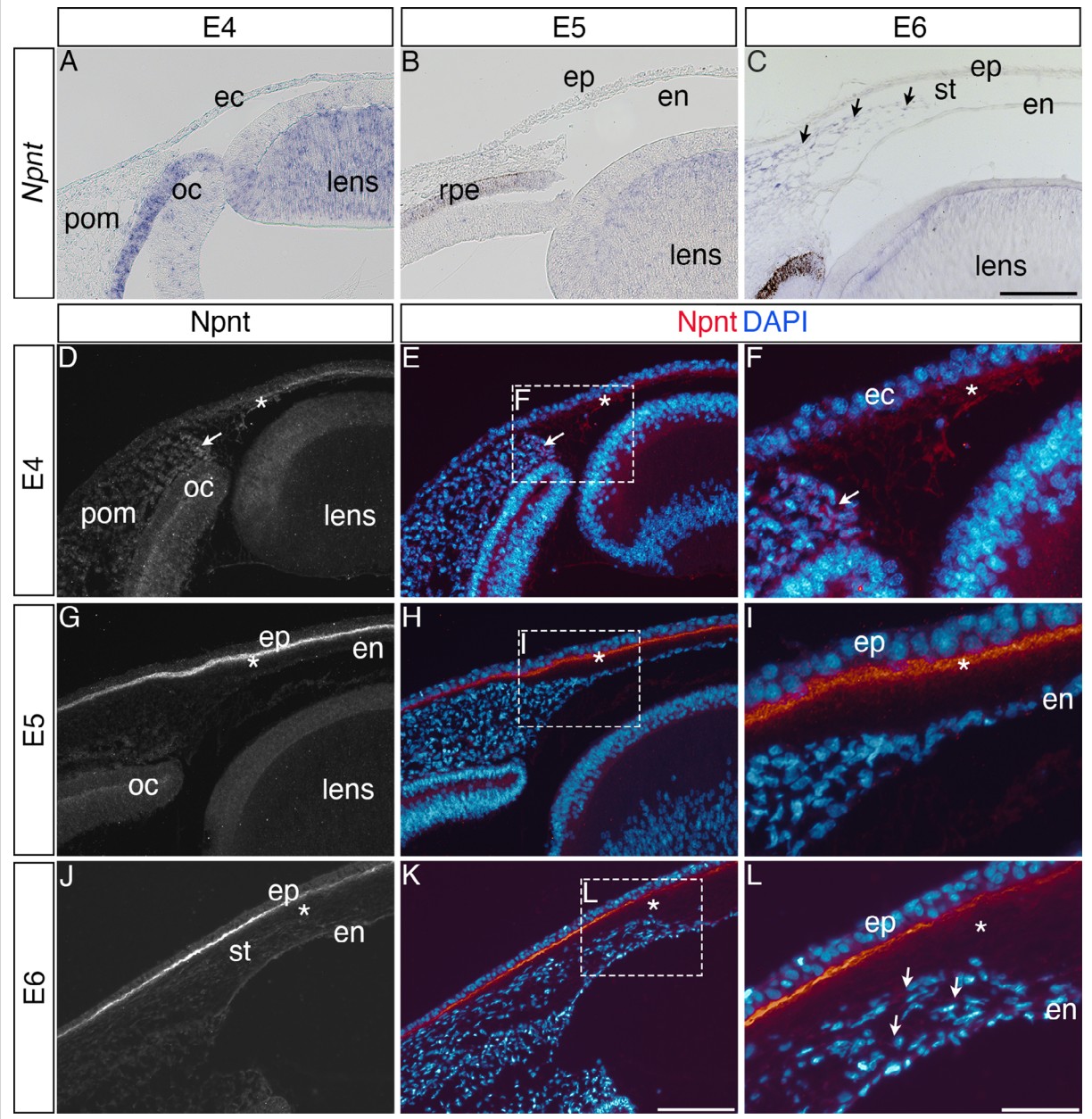

**Figure 1.** Nephronectin (Npnt) expression during early ocular development. Expression of Npnt mRNA and protein was examined via section in situ hybridization (**A–C**) or immunohistochemistry (**D–L**). (**A, B**) *Npnt* expression in the retina pigment epithelium layer of the optic cup and region of the presumptive lens fiber cells at embryonic day (E)4 and E5. (**C**) Initial expression of *Npnt* by periocular neural crest (pNC) is observed during the second wave of migration into the stroma (black arrows). (**D–F**) At E4, Npnt protein was detected in the optic cup, lens epithelium, in the periocular mesenchyme proximal to the presumptive cornea region (arrow), and in the matrix of the primary stroma (asterisk). (**G–I**) At E5, vivid expression of Npnt protein is localized in the primary stroma adjacent to the corneal epithelium and diffusely throughout the primary stroma (asterisk), and persists at low levels in the optic cup and lens epithelium. At E6, vivid expression of Npnt protein persists in the primary stroma adjacent to the corneal epithelium and it remains diffusely expressed throughout the primary stroma (asterisk). At this time, low expression of Npnt protein is also observed in the migratory periocular neural crest cells invading the primary stroma (arrows in **L**). ec, ectoderm; oc, optic cup; pom, periocular mesenchyme; rpe, retinal pigment epithelium; st, stroma; en, corneal endothelium; ep, corneal epithelium. Scale bars: 100 µm.

The online version of this article includes the following figure supplement(s) for figure 1:

**Figure supplement 1.** Expression of nephronectin (Npnt) transcripts and protein during late stages of development of the chick cornea.

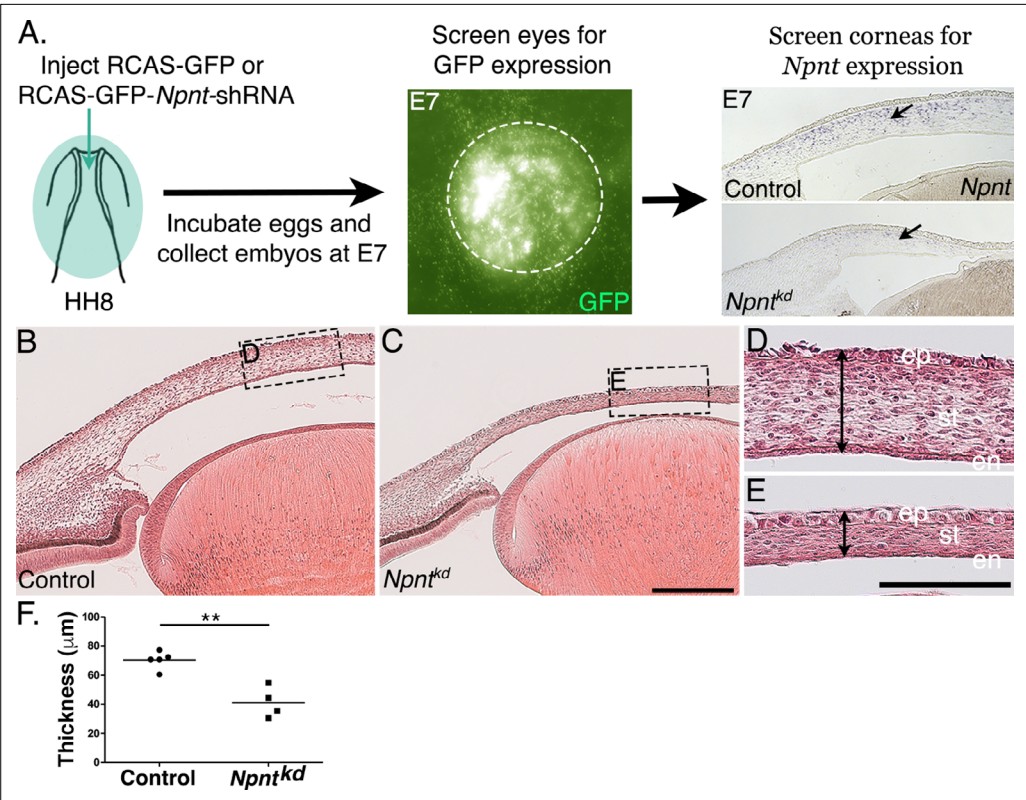

**Figure 2.** Corneal thickness is reduced in *Npnt*^kd^ corneas. (**A**) Schematic of in ovo injection of viral constructs (green) to cover the anterior region of stage 8 chick embryo. Following 7 days of incubation, embryos were screened for GFP expression in the anterior eye region. Knockdown was verified by section in situ hybridization, which revealed reduced expression of *Npnt* in *Npnt*^kd^ cornea compared with control. (**B–E**) Hematoxylin and eosin staining showing control (**B, D**) and thinner *Npnt*^kd^ corneas (**C, E**). Statistical analysis on measurements taken from (N = 5 control and N = 4 *Npnt*^kd^ corneas) revealed (**F**) significant reduction in thickness of *Npnt*^kd^ corneas. **p<0.01. ep, corneal epithelium; st, stroma; en, corneal endothelium;. Scale bars: 100 µm (**B, C**), 100 µm (**D, E**).

The online version of this article includes the following figure supplement(s) for figure 2:

**Figure supplement 1.** Diagram of replication-competent ASLV long terminal repeat with a splice acceptor (RCAS) vectors used for expression of either (**A**) GFP alone (control) or GFP together with inserts of shRNA or full-length nephronectin (Npnt).

(*Figure 2C*) compared to control (N = 5; *Figure 2B*). Measurements taken in the mid-corneal regions (*Figure 2D and E*; arrowheads) showed significant reduction in corneal thickness in *Npnt*^kd^ corneas (*Figure 2F*). These data suggest that Npnt plays a critical role during early development of the cornea.

## Itgα8 is the receptor for Npnt during cornea development and plays a role in pNC migration

Next, we reasoned that because knockdown of *Npnt* causes a significant reduction in corneal thickness, one or more of its receptors should be expressed by the migratory pNC during cornea development. In this study, we focused on α8β1 because of its strong affinity for Npnt (*Brandenberger et al., 2001*). Although α8β1 was observed to promote spreading of trunk neural crest in vitro (*Testaz et al., 1999*), little is known about its expression and function in the cranial neural crest. Our analysis by in situ hybridization revealed that *Itgα8* was expressed in the leading edge of the periocular mesenchyme prior to pNC migration into the primary stroma of the nascent cornea (*Figure 3A*, arrow). Subsequently, *Itgα8* was maintained in the periocular mesenchyme but also expressed in the corneal endothelium at E5 (*Figure 3B*, arrows) and stroma at E6 (*Figure 3C*).

Given the striking expression of *Itgα8* by the migratory pNC, we first assessed the potential for Itgα8-Npnt signaling in vitro using explanted periocular mesenchyme from the leading edge adjacent

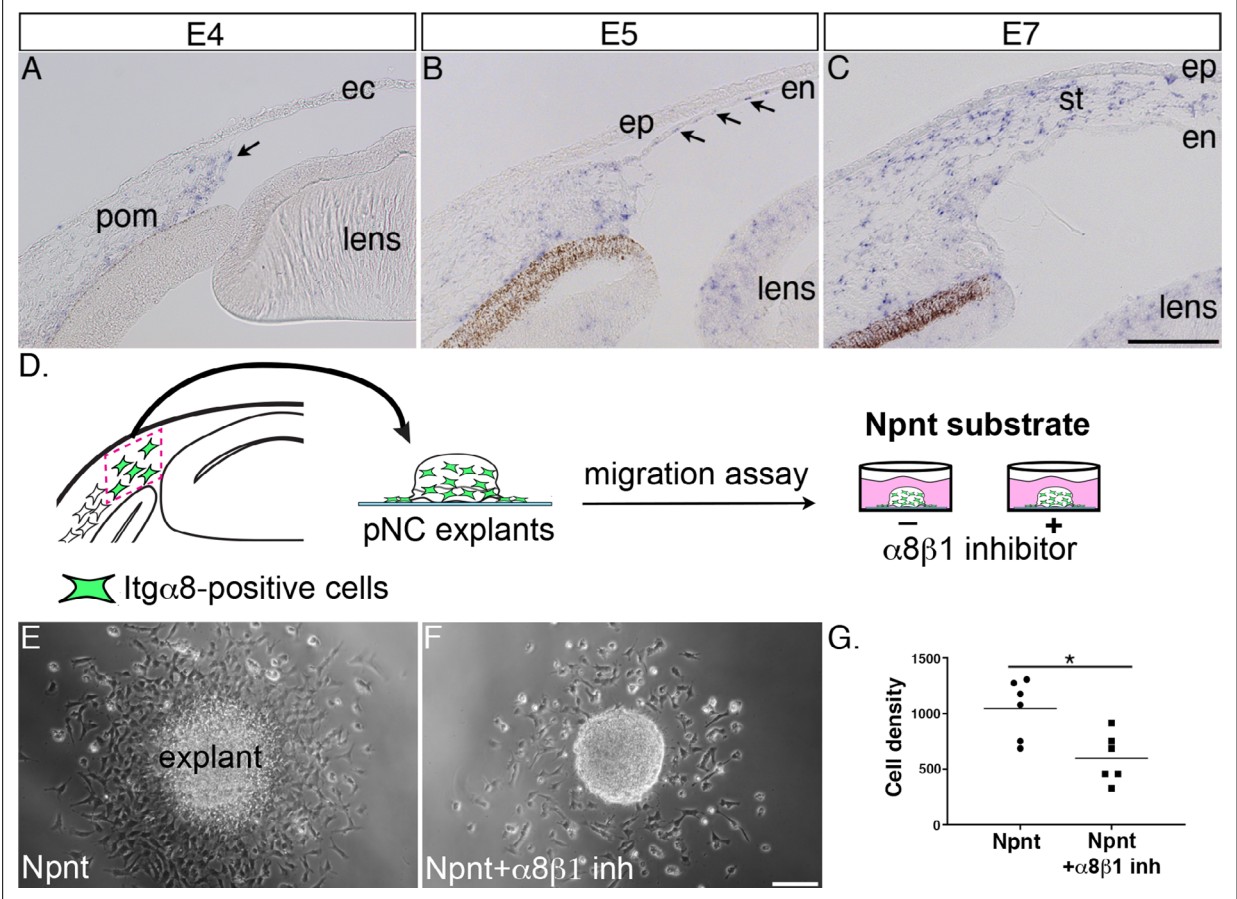

**Figure 3.** Itgα8 is expressed periocular neural crest (pNC) during cornea development and plays a role in cell migration. (**A**) Expression of *Itgα8* is observed in pNC prior to their migration into the cornea at embryonic day (E)4 (arrow). (**B**) *Itgα8* is subsequently expressed in the corneal endothelium at E5 (arrows) and (**C**) the migratory pNC in the corneal stroma at E6. (**D**) Schematic showing the isolation of periocular mesenchyme used for generating pNC explants for in vitro migration on Npnt-coated substrate in the presence or absence of α8β1 inhibitor. (**E**) Explant cultured on Npnt substrate showing robust cell migration after 12 hr. (**F**) Explant cultured on Npnt substrate in the presence of α8β1 inhibitor showing fewer cell migration after 12 hr. (**G**) Statistical analysis performed on N = 6 explants on Npnt substrate and N = 6 explants on Npnt substrate plus inhibitor revealed significant reduction in cell density of migratory cells in the presence of the inhibitor. *p<0.05. ec, ectoderm; pom, periocular mesenchyme; en, corneal endothelium; ep, corneal epithelium; st, stroma. Scale bars: 100 µm.

The online version of this article includes the following video for figure 3:

**Figure 3—video 1.** Migration of periocular neural crest (pNC) from mesenchyme explant on nephronectin (Npnt)-coated substrate.
https://elifesciences.org/articles/74307/figures#fig3video1

**Figure 3—video 2.** Migration of periocular neural crest (pNC) from mesenchyme explant on nephronectin (Npnt)-coated substrate in the presence of α8β1 inhibitor.
https://elifesciences.org/articles/74307/figures#fig3video2

to the presumptive cornea (*Figure 3A and D*). Mesenchyme explants were cultured on slides coated with Npnt in the presence or absence of a peptide inhibitor previously shown to specifically inhibit binding of α8β1 with Npnt (*Figure 3D*; *Sato et al., 2009*). In the absence of the inhibitor, explants attached to the Npnt-coated slides and numerous migratory cells formed a halo around the explant within 12–17 hr of incubation (*Figure 3E*, *Figure 3—video 1*). In contrast, we observed that treatment with the α8β1 inhibitor significantly decreased the density of migratory cells from the explant (*Figure 3F and G*, *Figure 3—video 2*). These results indicate that α8β1 functions as a receptor for Npnt signaling in the presumptive corneal pNC, and that it is required for their migration.

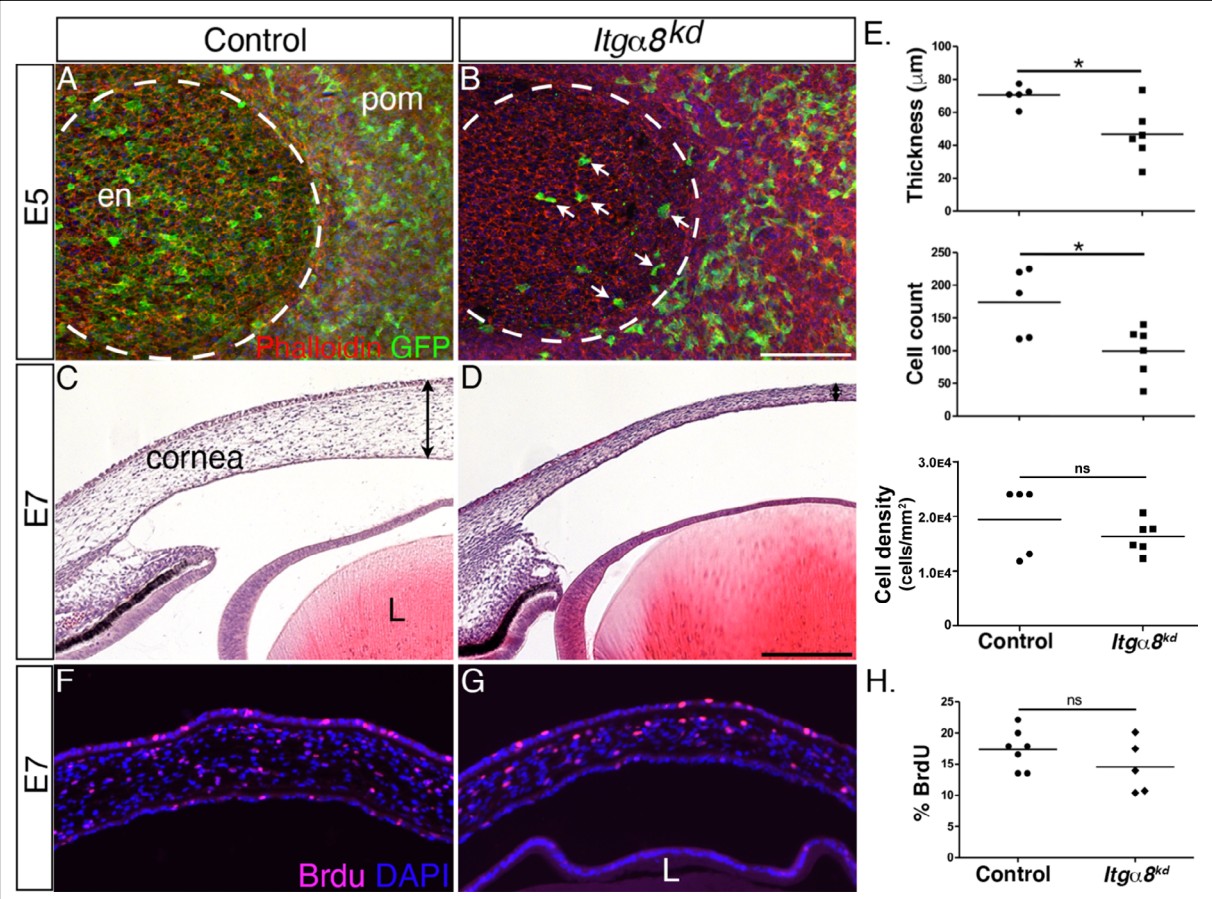

**Figure 4.** Knockdown of Itgα8 reduces periocular neural crest (pNC) migration and results in reduced corneal thickness. (**A, B**) Whole-mount embryonic day (E)5 anterior eyes immunostained for GFP and counterstained with phalloidin to reveal cell membranes. (**A**) Control eye showing robust expression of GFP by pNC cells that migrated into the corneal region to form the endothelial layer. (**B**) Itgα8$^{kd}$ eye showing that relatively fewer GFP cells migrated into the cornea (arrows). Dotted lines demark the boundary between the cornea and periocular mesenchyme. (**C, D**) Hematoxylin and eosin staining of E7 corneal section showing (**C**) normal corneal thickness in control and (**D**) reduced corneal thickness in Itgα8$^{kd}$ embryos. Double-sided arrows indicate corneal thickness. (**E**) Statistical analysis of measurements taken from N = 5 control and N = 6 Itgα8$^{kd}$ corneas revealed significant reduction in thickness and cell count, and no difference in cell density in Itgα8$^{kd}$ corneas, *p<0.05. (**F, G**) Bromodeoxyuridine (BrdU) immunofluorescent analysis of cell proliferation in E7 corneal sections. Quantification of BrdU-positive cells in the corneal stroma was performed by normalizing to the total number of DAPI-positive cells. (**H**) Statistical analysis from N = 7 control and N = 5 Itgα8$^{kd}$ revealed no difference between control and Itgα8$^{kd}$ corneas. ns, not significant; en, corneal endothelium; pom, periocular mesenchyme; L, lens. Scale bars: 100 μm.

The online version of this article includes the following figure supplement(s) for figure 4:

**Figure supplement 1.** Validation of Itgα8 knockdown in vivo.

## Knockdown of *Itgα8* disrupts corneal thickness during development

Given that *Itgα8*-expressing pNC appear to be in direct contact with Npnt secreted into the primary stroma, we wanted to characterize the role of Itgα8 during pNC migration. For this analysis, we generated and tested several RCAS-GFP-Itgα8-shRNA constructs and validated one that showed highest knockdown of *Itgα8* in DF-1 cells (*Figure 2—figure supplement 1C*) and in vivo (*Figure 4—figure supplement 1*). As a first step, we analyzed the effect of *Itgα8* knockdown on the first wave of pNC migration that forms the corneal endothelium at E5. We found that at this time point relatively fewer GFP-positive cells expressing the *Itgα8$^{kd}$* construct appeared to migrate into the cornea (*Figure 4B*, arrows) compared to RCAS-GFP control, which showed robust occupation of GFP-positive cells in the corneal endothelium (*Figure 4A*). Despite the attenuated migration of pNC expressing the *Itgα8$^{kd}$* construct during the first wave, we did not observe defects in the corneal endothelium. One possibility is that pNC, which do not endogenously express Itgα8 (*Figure 3A*), may also contribute to the corneal endothelium, albeit at a lower level, but they are able to compensate for the *Itgα8*

knockdown, resulting in the formation of a normal endothelial layer. Nonetheless, histological analysis at E7 revealed reduction in corneal thickness following *Itgα8* knockdown (*Figure 4D*) compared to control corneas (*Figure 4C*). Measurements and quantification of cells in the mid-corneal regions showed an overall significant reduction in corneal thickness and cell number, but no difference in cell density between *Itgα8^kd* and control corneas (*Figure 4E*). Furthermore, we performed a bromodeoxyuridine (BrdU) assay and observed no significant differences in cell proliferation between *Itgα8^kd* (*Figure 4G*) and control corneas (*Figure 4F*). As indicated in our analysis at E5, it is likely that non-Itgα8-expressing pNC may compensate during the second wave of pNC migration, but not to the extent that abrogates the corneal thinning defect, possibly due to the relatively large number of cells required for the formation of the stroma. These results indicate that Itgα8 is required for pNC migration into the cornea. Given that the corneal thinning defect following knockdown of *Itgα8* correlates with *Npnt* knockdown, our results suggest that Npnt-Itgα8 signaling plays an important role during pNC migration into the cornea.

## Overexpression of *Npnt* causes corneal thickening

Given that knockdown of both *Npnt* and *Itgα8* resulted in corneal thinning, combined with our observation that *Npnt* is expressed during the second wave of migration at E6, we investigated whether overexpression of *Npnt* affects cornea development. We generated and tested viral constructs containing the full-length Npnt gene (*Npnt^oe*) in DF1 cells (*Figure 2—figure supplement 1C*). Control and *Npnt^oe* constructs were injected in HH7-8 embryos as described (*Figure 2*), and the corneas were collected for analysis at E7, E9, and E15. First, we confirmed the overexpression of Npnt mRNA and protein in the corneas. At E9, expression of *Npnt* was localized in the corneal stroma during normal development (*Figure 5A*). However, *Npnt^oe* corneas showed robust expression of *Npnt* in the stroma, as well as ectopic expression in the corneal epithelium and endothelium, and the lens (*Figure 5D*). At this stage, strong staining for Npnt protein was only observed in the basement membrane of the corneal epithelium and low diffuse staining in the stroma of control corneas (*Figure 5B*). In contrast, the *Npnt^oe* corneas showed strong ectopic staining for Npnt in the corneal epithelium and endothelium, and a substantial increase in protein expression in the stroma compared to the control (*Figure 5E*).

Next, we performed histological analysis on E7, E9, and E15 corneas to determine whether there were any morphological differences between the control and *Npnt^oe* corneas. Our analysis did not reveal any significant differences in corneal thickness and cell number at E7 (*Figure 5G and H*). By contrast, cross-sections of E9 *Npnt^oe* corneas revealed significant thickening and increase in the cell number (*Figure 5F–H*) compared with control (*Figure 5C, G and H*). Similar increases in corneal thickness and cell number were observed at E15 (*Figure 5G and H*). Given that the keratocyte density is higher in the anterior stroma than the posterior stroma (*Patel et al., 2001*; *Berlau et al., 2002*), we quantified the cell densities in these regions. Overall, we observed relatively higher cell densities in the anterior stroma of E7, E9, and E15 corneas and there were no significant differences between *Npnt^oe* and control corneas (*Figure 5I*). Posterior cell density was not affected in E7 *Npnt^oe* corneas, but there was significant reduction at E9 and E15.

To determine whether elevated cell proliferation was a contributing factor to the increased corneal thickness in *Npnt^oe*, we examined corneas at E7, E9, and E15 by performing BrdU labeling and immunofluorescent detection. Overall, we observed cell proliferation in all corneal layers, and there was a decreasing trend in the percentage of labeled cells in the stroma as development progressed from E7 to E15 (*Figure 5J and K*). However, our results revealed no significant increase in cell proliferation at E7 and E15. Surprisingly, there was a significant reduction in cell proliferation in E9 *Npnt^oe* corneas despite their increase in thickness (*Figure 5K*). Combined, our data indicate that overexpression of *Npnt* increases transcript and protein expression in the cornea, which results in increased cell number in the stroma and corneal thickness. Since these increases were not correlated with increased cell proliferation, our data suggest that the changes in corneal thickness are due to augmented pNC migration into the cornea caused by excessive expression of Npnt.

## The RGD domain mediates Npnt signaling during pNC migration into the cornea

Npnt mediates signal transduction in development and cancer by binding to various receptors through the EGF-like and RGD domains (*Arai et al., 2017*; *Kahai et al., 2010*; *Kuek et al., 2016*; *Linton et al.,*

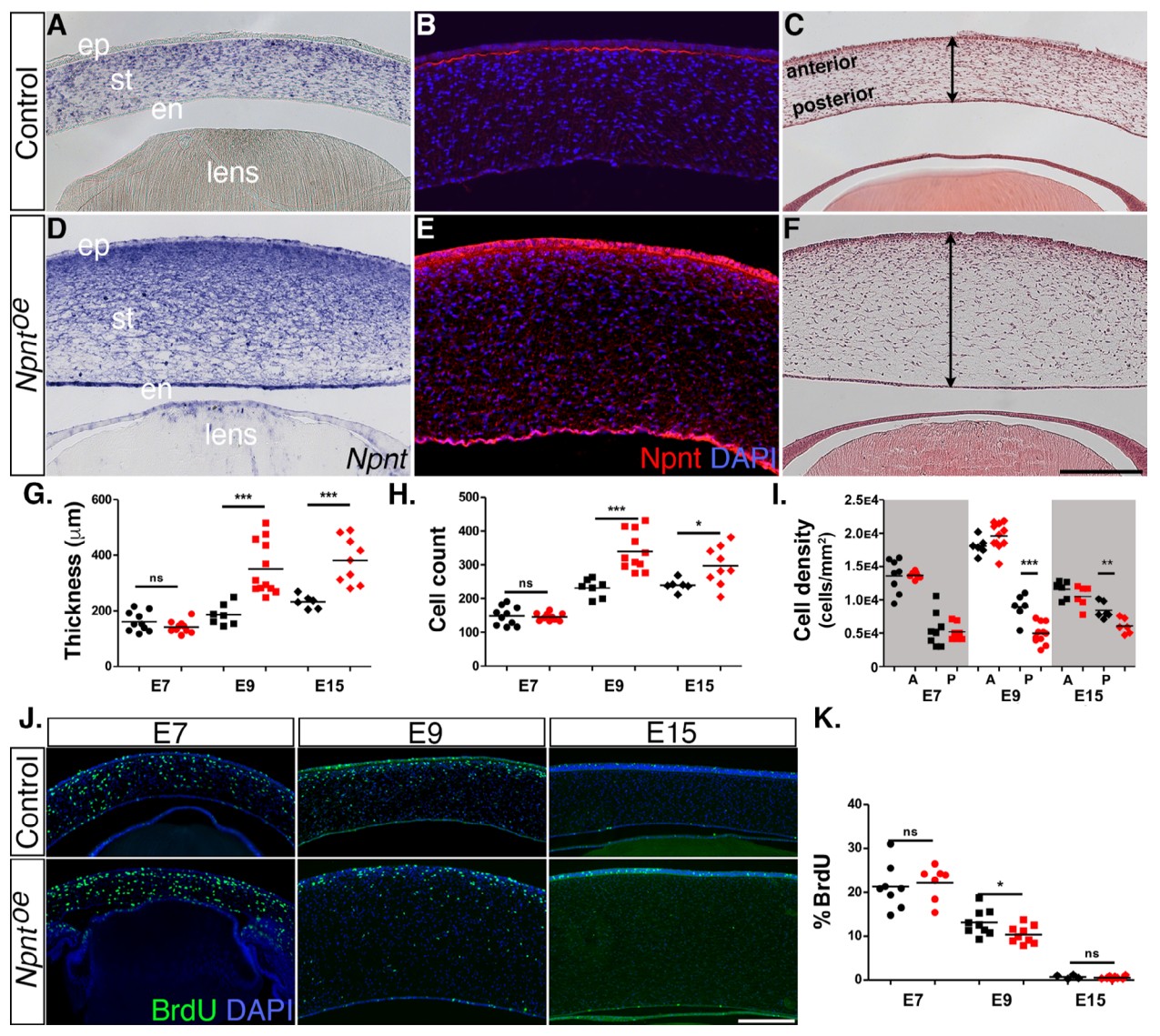

**Figure 5.** Effects of overexpression of nephronectin (Npnt) during corneal development. Embryos were injected with RCAS virus expressing GFP alone (control) or GFP and the full-length Npnt protein, and corneas were analyzed at the developmental stages indicated. (A–F) Representative corneal sections from embryonic day (E)9 control (A–C) and $Npnt^{oe}$ corneas (D–F) showing levels of Npnt transcript (A, D) and protein (B, E) expression, and hematoxylin and eosin staining indicating corneal thickness (double-sided arrows, C, F). (G, H) Measurements for corneal thickness and cell counts were taken from E7, N = 10 control, N = 10 $Npnt^{oe}$; E9; N = 7 control, N = 12 $Npnt^{oe}$; E15; N = 6 control, N = 9 $Npnt^{oe}$. Bar graphs show no difference at E7, but a significant increase at E9 and E15 in corneal thickness (G) and corneal cells (H). (I) Cell densities were determined from E7, N = 8 control, N = 7 $Npnt^{oe}$; E9; N = 6 control, N = 11 $Npnt^{oe}$; E15; N = 7 control, N = 6 $Npnt^{oe}$. Bar graph shows that there were no differences at E7 and anterior cell densities at E9 and E15, but the posterior cell densities were significantly decreased. (J, K) Bromodeoxyuridine (BrdU) analysis and quantification of cell proliferation in corneal sections taken from E7, N = 8 control, N = 7 $Npnt^{oe}$; E9; N = 9 control, N = 9 $Npnt^{oe}$; E15; N = 7 control, N = 6 $Npnt^{oe}$. No significant differences were observed at E7 and E15, but there was a significant reduction at E9. ns, not significant; *p<0.05; **p<0.01; ***p<0.001. ep, corneal epithelium; st, stroma; en, corneal endothelium. Scale bars: 100 µm.

*2007*). The marked reductions in corneal thickness observed following knockdown of both *Npnt* and *Itgα8* (**Figures 2 and 3**) suggest that Npnt functions via the RGD domain during pNC migration. Given that overexpression of the full-length Npnt protein caused corneal thickening, we generated an Npnt-RAE version of the protein in which the RGD domain was mutated by substitution with RAE sequence, and an Npnt-EGF version in which both the RGD and MAM domains were truncated (**Figure 6A**). Constructs were injected in HH7-8 embryos, and E9 corneas were collected, sectioned, and analyzed following DAPI staining. Contrary to our observations following overexpression of

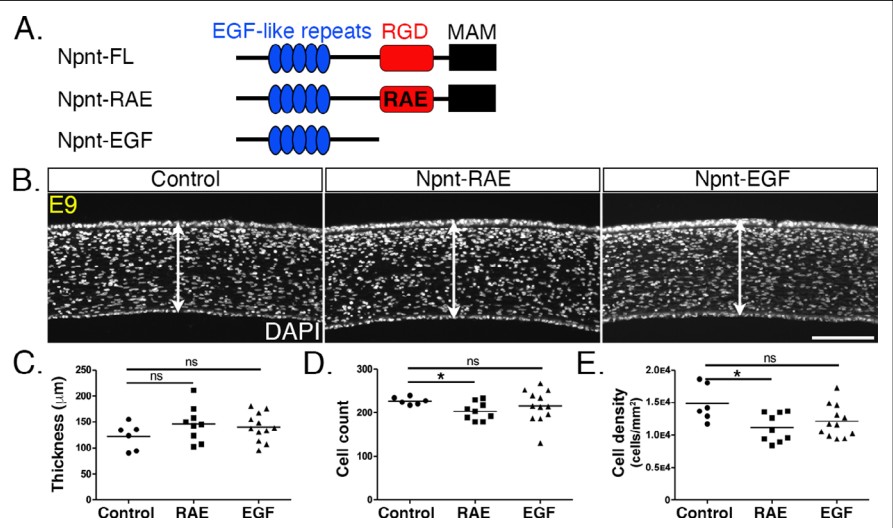

**Figure 6.** Overexpression of versions of nephronectin (Npnt) with either mutant or truncated RGD domains does not increase corneal thickness. (**A**) Schematic showing the full-length Npnt, Npnt with mutated RGD to RAE sequence, and the truncated version containing only the epidermal growth factor (EGF) domain. (**B**) Representative sections of embryonic day (E)9 corneas showing corneal thicknesses (double-sided arrows) following overexpression of control, RGD mutant, and truncated versions of Npnt. (**C–E**) Quantification of measurements taken from N = 5 control, N = 9 RAE, and N = 12 EGF showing (**C**) no significant differences in corneal thickness. (**D, E**) Significant reduction in cell count and density in RGD mutant, but no difference in the truncated version. ns, not significant; *p<0.05. Scale bar: 100 μm.

the full-length protein (*Figure 5*), neither the Npnt-RAE nor Npnt-EGF versions caused a significant change in corneal thickness compared with the control (*Figure 6B and C*). Interestingly, there was a significant reduction in cell count and density in Npnt-RAE corneas, although no difference in these values was observed for Npnt-EGF (*Figure 6D and E*). The absence of corneal thickening following overexpression of either Npnt-RAE or Npnt-EGF constructs further suggests that the RGD domain plays an essential role in this process. Our results also indicate that the EGF-like domain does not play an essential role during pNC migration into the cornea.

## Npnt-Itgα8 signaling mediates pNC migration via a Rho-mediated mechanism

Previous studies have suggested that the Rho-associated kinase (ROCK) pathway is involved with integrin α8β1 modulation of actin stress fibers in vascular smooth muscles and intestinal crypt cells (*Benoit et al., 2009*; *Zargham et al., 2007*). To explore the mechanism by which Npnt-Itgα8 signaling modulates pNC migration, we performed explant culture experiments for 24 hr on Npnt substrate as control (*Figure 7A*) and compared them to explants cultured in the presence of Npnt combined with various inhibitors. Given that FAK play a role in integrin-mediated migration (*Ilić et al., 1995*; *Parsons, 2003*), we first wanted to verify if Npnt-Itgα8-induced migration is mediated through the modulation of focal adhesions. We compared explants cultured on Npnt with the inhibitor for α8β1 or FAK and observed that similar to the α8β1 inhibitor (*Figure 7—figure supplement 1C and D*), inhibition of FAK reduced pNC migration from the explant but also severely attenuated cell attachment (*Figure 7—figure supplement 1E and F*).

Next, to examine if Npnt-Itgα8 signaling induces assembly of the cytoskeletal machinery required for pNC migration via the Rho pathway, we treated the explants with the ROCK inhibitor Y27632. Under these conditions, the cells dissociated from the explant, attached to the Npnt substrate, and transformed into a rounded phenotype with significantly increased surface area compared to Npnt alone (*Figure 7A–C*). In addition, our analysis indicated that the ROCK inhibitor significantly decreased the density of migratory cells (*Figure 7D*). Given that the cells covered a wider area, it is possible that the decrease in cell density could be due to the increased surface area or decreased cell migration. We therefore tested for linear patterns of cell dispersion from the explant. Our results revealed that in

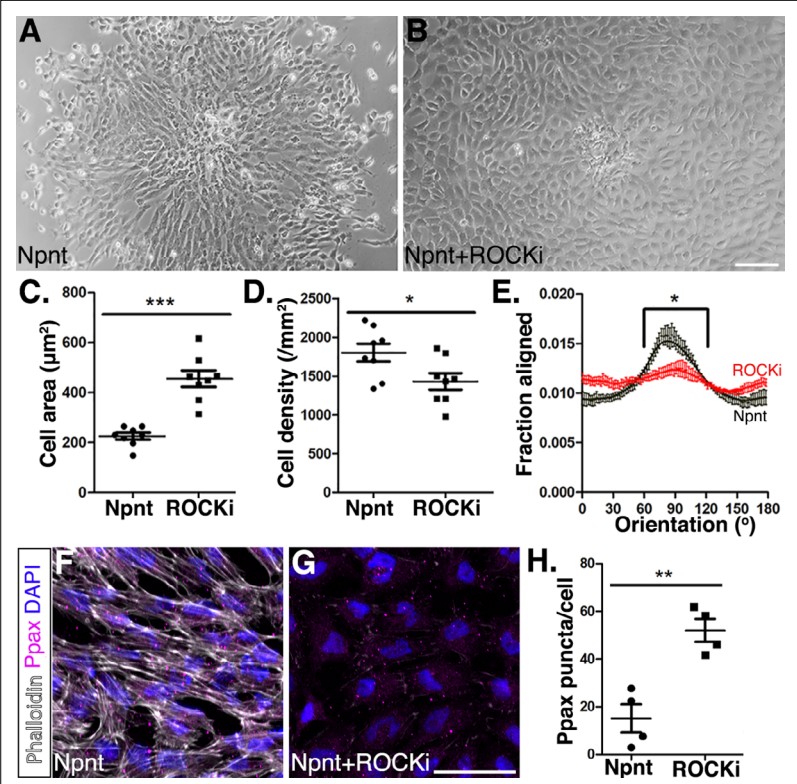

**Figure 7.** Inhibition of Rho-kinase attenuates periocular neural crest (pNC) migration on nephronectin (Npnt). (**A–D**) Comparisons of pNC migration from explants cultured for 24 hr on either (**A**) Npnt substrate alone or in the presence of (**B**) the ROCK inhibitor Y27632. (**C, D**) Quantification of cell area taken from N = 8 Npnt and N = 8 Npnt plus ROCK inhibitor showing significant reduction in (**C**) cell area and (**D**) cell migration in the presence of ROCK inhibitor. (**E**) Quantification of cell orientation from N = 6 Npnt and N = 6 Npnt plus ROCK inhibitor explants showing significant difference in the fraction aligned between 90° ± 30°. (**F, G**) Cells stained for actin (phalloidin) and focal adhesion (pY118 paxillin) showing (**F**) formation of actin stress fibers and focal adhesions by pNC migrating on Npnt. (**G**) Substantial decrease in actin stress fibers in the presence of the ROCK inhibitor. (**H**) Quantification taken from N = 4 Npnt and N = 4 ROCK inhibitor explants, showing significant increase in the number of pY118 paxillin-positive puncta. *$p < 0.05$; **$p < 0.01$. Scale bars: (**A, B**) 100 µm; (**F, G**) 50 µm.

The online version of this article includes the following figure supplement(s) for figure 7:

**Figure supplement 1.** Analysis of periocular neural crest (pNC) migration on nephronectin (Npnt) substrate in the presence of inhibitors of components of the migratory signaling pathway.

the presence of the ROCK inhibitor cells exhibited a significantly reduced preferred orientation (90° ± 30°) compared to Npnt alone (*Figure 7E*). Thus, in the presence of the ROCK inhibitor, the pNC spread out on the Npnt substrate with limited directional migration. To test if loss of directionality was caused by changes to the cytoskeletal structure, we examined the cytoskeletal machinery of the cells using phalloidin staining and focal adhesions by immunostaining for pY118 paxillin, which provides a docking site for assembly of the actin fibers that is required for cell adhesion, spreading, and migration (*Nakamura et al., 2000*; *Turner, 2000*). As previously reported (*Katoh et al., 2001*; *Masamune et al., 2003*; *Takamura et al., 2001*), the ROCK inhibitor attenuated the formation of actin stress fibers in pNC as indicated by the substantial reduction in the intensity of phalloidin staining (*Figure 7F and G*). Cells in both conditions stained positive for pY118 paxillin, but the number of puncta per cell significantly increased in the presence of the ROCK inhibitor. This result indicates that disruption of Rho kinase prevents pNC from forming the actin stress fibers and that the cells adhere to the Npnt substrate using multiple foci. Collectively, these data show that Npnt/Itgα8 signaling activates FAK/paxillin for adhesion and that the Rho signaling pathway plays an important role during pNC migration on Npnt.

**Figure 8.** Dynamics of periocular neural crest (pNC) response to the extracellular matrix (ECM) in the context of the expression of nephronectin (Npnt) and fibronectin (Fn) during corneal development. (**A**) Cross-section of embryonic day (E)5 eye immunostained for Npnt and Fn. Npnt appears in an increasing gradient from the periocular region into the cornea, whereas Fn stains both the periocular mesenchyme and cornea. (**B**) All pNC respond to Fn via expression of α5β1, but a subpopulation of pNC that reside in the region adjacent to the presumptive cornea express both α5β1 and α8β1, and become competent to also read the additional gradient of Npnt in the ECM, thus migrating into the corneal region. oc, optic cup; ps, primary stroma; en, corneal endothelium.

## Discussion

Very little is known about the function of the corneal ECM during early development. Here, we focus on Npnt, which we recently found in our RNA-seq study (*Bi and Lwigale, 2019*) to be upregulated during early development of the chick cornea. Specifically, we have identified novel expression of Npnt and its Itgα8 receptor at critical time points during pNC migration into the nascent cornea. Our data reveal that disruption of Npnt/Itgα8 signaling either in ovo or in vitro perturbs pNC migration, which subsequently results in corneal thickness defects, but formation of the cellular layers is not affected. Our model (*Figure 8*) provides a context in which Npnt functions in the presence of Fn during corneal development. Fn is a well-known substrate for neural crest cell migration (*Alfandari et al., 2003*; *Bronner-Fraser, 1986*; *Newgreen and Thiery, 1980*) that is robustly expressed in the periocular mesenchyme and cornea (*Doane et al., 1996*; *Kurkinen et al., 1979*). In comparison, Npnt is expressed in an increasing gradient from the edge of the periocular region towards the cornea (*Figure 8A*). We posit that all pNC cells express α5β1 and previously showed that they robustly migrate on Fn substrate in vitro (*Lwigale and Bronner-Fraser, 2009*). However, in vivo, it is most likely that the pNC, which express both α5β1 and α8β1, and thus can respond to Npnt and Fn, migrate into the cornea. Based on this study, we conclude that Npnt/Itgα8 signaling plays an essential role in pNC migration during corneal development.

### Npnt and Itgα8 in the cornea

While Npnt is associated with kidney, teeth, and bone development (*Arai et al., 2017*; *Kuek et al., 2016*; *Linton et al., 2007*), only one reference was made to its expression in the mouse lens during ocular development (*Brandenberger et al., 2001*). First, we confirmed that *Npnt* is expressed in the migratory pNC and showed that it is also localized in the optic cup and lens vesicle. Our immunohistochemistry analysis revealed that Npnt protein staining consistently corresponded with the mRNA expression during pNC migration into the developing cornea. In addition, we observed that Npnt was localized in the acellular primary stroma, suggesting that prior to its expression by the migratory pNC in the cornea, the optic cup and lens vesicle are the primary sources of Npnt in the nascent corneal ECM. The link between the primary stroma and the presumptive corneal epithelium and endothelium was established in classical studies, which showed that the lens and corneal endothelium induce the corneal epithelium to synthesize collagen and glycosaminoglycans into the underlying space (*Fitch et al., 1994*; *Hay et al., 1979*; *Hay and Revel, 1969*; *Toole and Trelstad, 1971*). Our expression analyses suggest that the primary stroma sequesters Npnt initially secreted by the optic cup and the lens, and exposes it to pNC during early corneal development.

Itgα8 is expressed in the spinal cord, optic nerve, retina, urogenital and digestive systems, and in the head epidermis (*Bossy et al., 1991*; *Ogawa et al., 2018*). α8β1 is known to modulate epithelial–mesenchymal interactions during kidney and pharyngeal development (*Müller et al., 1997*; *Talbot*

et al., 2016) and promote migration of mesangial and smooth muscle cells (*Bieritz et al., 2003*; *Zargham and Thibault, 2006*). We report for the first time that *Itgα8* is expressed by pNC during ocular development. Since Itgα8 heterodimerizes with Itgβ1 subunit (*Bossy et al., 1991*), which is robustly expressed in the periocular mesenchyme and migratory pNC (*Doane and Birk, 1994*), we infer that α8β1 signaling functions during corneal development in response to Npnt secreted in the primary stroma. Following delamination from the neural tube, cranial neural crest migrate in response to Fn and laminin (*Duband and Thiery, 1982*; *Newgreen and Thiery, 1980*; *Sternberg and Kimber, 1986*) via integrin receptors such as α1β1, α4β1, α5β1, and αVβ1 (*Alfandari et al., 2003*; *Delannet et al., 1994*; *Desban and Duband, 1997*; *Kil et al., 1996*; *Lallier et al., 1994*). Although neural crest cells are initially highly migratory, they remain relatively stationary upon their localization in the periocular region. Given that *Itgα8* expression coincides with the onset of pNC migration, our results indicate its potential role in their ingression into the developing cornea.

## Npnt promotes pNC migration via RGD domain, Itgα8, and the Rho/Rock pathway

The three major factors that contribute to corneal thickness during development are pNC migration, cell proliferation, and synthesis of the secondary stroma by the keratocytes. We showed that disruption of *Npnt* expression in the anterior ocular tissues via RCAS-mediated knockdown resulted in decreased corneal thickness. In addition to functioning in epithelial–mesenchymal interactions, Npnt has been shown to promote migration in various tissues, including vascular endothelial cells during osteogenesis (*Kuek et al., 2016*), infiltration of immune cells into the liver (*Hong et al., 2020*; *Inagaki et al., 2013*), and cancer metastasis (*Magnussen et al., 2020*; *Mei et al., 2020*; *Wang et al., 2018*). Given that Npnt is localized in the primary stroma during corneal development, we hypothesized that it may play a potential role in pNC migration. In agreement with our hypothesis, our in vitro migration assays confirmed that robust pNC migration from mesenchyme explant occurred on Npnt substrate but was abrogated in the presence of the Itgα8 inhibitor. We also observed that knockdown of *Itgα8* reduced pNC migration into the cornea and phenocopied the reduction of corneal thickness observed following *Npnt* knockdown. In addition, we found that reduction in corneal thickness was accompanied by a decrease in cell count, but cell density and proliferation were not affected, suggesting that Npnt/Itgα8 signaling mediates pNC migration during corneal development. Given that the primary function of keratocytes is to synthesize the corneal ECM comprising collagens and proteoglycans, and represents approximately 90% of the corneal thickness (*Fini, 1999*; *Funderburgh et al., 2003*; *Kao, 2010*), our observation that there was no change in stromal cell density raises a possibility that matrix synthesis by the pNC that differentiated into keratocytes is not a major contributing factor to the reduction in corneal thickness. However, additional studies may be required to determine how the disruption in corneal thickness observed in this study may affect the corneal ECM and transparency at later stages of development.

Our overexpression studies showed the opposite effect whereby the full-length construct caused increased corneal thickness at E9. We also found that increased corneal thickness persisted at E15 when Npnt appears to be downregulated in the cornea. Surprisingly, the E7 corneas were not affected in the overexpression studies. This could be caused by limited pNC migration potential due to their expression of Neuropilin1, which prevents them from entering the corneal environment that contains a repulsive Semaphorin3A signal (*Lwigale and Bronner-Fraser, 2009*). As observed in the knockdown studies, there were no differences in cell proliferation between control and *Npnt*^oe corneas, suggesting that there was an increase in pNC migration. The mutant Npnt-RAE construct overexpresses approximately the same-size protein as the endogenous Npnt, while maintaining its MAM domain to support ECM–ECM interactions, but it did not impact cornea thickness. This indicates that the expression of high levels of protein does not affect cornea thickness, further implicating that Npnt functions in directing cell migration specifically through its RGD sequence during early cornea development. The reduction in cell count and density could be attributed to the mutant RAE domain outcompeting the expression of the endogenous RGD by the pNC. Similarly, the truncated version of Npnt containing only the EGF-like repeats did not affect cornea thickness, further confirming that Npnt signals through the RGD domain and Itgα8 to modulate pNC migration into the cornea.

Furthermore, our in vitro culture experiments revealed that directed migration of pNC from the explant requires activation of FAK by α8β1 to promote focal adhesions. Previous studies have shown

that activation of the Rho/Rock pathway via integrin signaling drives actin remodeling that is necessary for cell adhesion and migration (*Clark et al., 1998*; *Cox et al., 2001*; *Price et al., 1998*). Our data also revealed that in the presence of the ROCK inhibitor pNC transformed into a rounded morphology in which assembly of the actin cytoskeletal machinery was disrupted and the number of focal adhesions increased. These results indicate a drastic change in migratory behavior that was less polarized from the explant, implying a potential involvement of the Rho/ROCK pathway in the directed migration pNC into the cornea. Given that pNC migrate on both Fn and Npnt in vitro, and that both α5β1 and α8β1 activate the Rho/Rock pathway (*Benoit et al., 2009*; *Danen et al., 2005*; *White et al., 2007*; *Zargham et al., 2007*), it is possible that Rho activity increases as the cells migrate from the periocular region into the matrix of the developing cornea due to the presence of additional cues from Npnt.

Precise coordination of multiple signals from surrounding ocular tissues orchestrates the spatio-temporal migration and differentiation of multipotent pNC during the formation of the avian corneal endothelium and stromal keratocytes. Disruptions in the sequence of these early events can lead to significant defects in corneal development. Our study provides the first characterization of Npnt/Itgα8 function in pNC that contribute to the cornea. Despite being surrounded by an ECM rich in Fn, pNC remain relatively immobile in the periocular region until some become competent to respond to Npnt cues via the expression of α8β1. This transformation may be crucial for the timely induction of directional migration of pNC towards the gradient of additional cues generated by Npnt in the primary stroma (*Figure 8A and B*), and therefore plays a vital role in segregating the cornea progenitors from the rest of the periocular mesenchyme. In this study, we focused on the role of Npnt/Itgα8 signaling during pNC migration. Our observation that the Npnt protein subsequently localizes to the basement membrane of the epithelial layer, combined with its pleiotropic functions during development and in disease (*Yamada and Kamijo, 2016*), implies that Npnt may have other functions at later stages of cornea development. One possibility is that Npnt mediates epithelial–mesenchymal interactions between the cornea epithelium and stroma by signaling through the EGF-like domain to EGFRs in the epithelial cells. Future studies will expand on these findings to determine whether the corneal thinning and thickening phenotypes observed in this study affect transparency, which develops at later stages. Elucidation of the link between Npnt signaling and corneal cell differentiation may provide useful insights for future therapeutic applications for wound healing and regeneration studies.

## Materials and methods

**Key resources table**

| Reagent type (species) or resource | Designation | Source or reference | Identifiers | Additional information |
|---|---|---|---|---|
| Transfected construct (*Gallus gallus*) | shRNA: Npnt | This paper | | See Materials and methods, section 'Production of RCAS virus' |
| Transfected construct (*G. gallus*) | shRNA: Itga8 | This paper | | See Materials and methods, 'Production of RCAS virus' |
| Biological sample (*G. gallus*) | Primary periocular neural crest | This paper | | See Materials and methods, section 'In vitro explant culture' |
| Biological sample (*G. gallus*) | DF-1 cells | ATCC | Cat# CRL12203 | Lot# 58217603, no mycoplasma contamination detected |
| Antibody | Anti-Npnt (rabbit polyclonal) | Biorbyt | Cat# orb221700; RRID:AB_2905548 | IF (1:100) |
| Antibody | Anti-GFP (mouse monoclonal) | Invitrogen | Cat# A-6455; RRID:AB_221570 | IF (1:500) |
| Antibody | Anti-fibronectin (mouse monoclonal) | DHSB | Cat# B3/D6; RRID:AB_2105970 | IF (1:30) |
| Antibody | Anti-phosphorylated paxillin (rabbit polyclonal) | Invitrogen | Cat# 44-722G; RRID:AB_2533733 | IF (1:200) |
| Antibody | Anti-BrdU (mouse monoclonal) | DHSB | Cat# G3G4; AB_2618097 | IF (1:30) |
| Chemical compound, drug | Bromodeoxyuridine | Sigma | Cat# B5002 | |

*Continued on next page*

*Continued*

| Reagent type (species) or resource | Designation | Source or reference | Identifiers | Additional information |
|---|---|---|---|---|
| Chemical compound, drug | Poly-D-lysine | Sigma | Cat# P6407 | |
| Recombinant DNA reagent | pSLAX13 | Addgene | Cat# CT#232 | |
| Recombinant DNA reagent | RCAS plasmid | *Hughes et al., 1987*; doi.org/10.1128/jvi.61.10.3004–3012.1987 | | |
| Recombinant RNA reagent | Integrin α8 RNA probe | This paper | PCR primers | *Supplementary file 1* |
| Recombinant RNA reagent | Nephronectin RNA probe | This paper | PCR primers | *Supplementary file 1* |
| Peptide, recombinant protein | α8β1 inhibitor | *Sato et al., 2009*; doi.org/10.1074/jbc.M900200200 | GenScript | |
| Peptide, recombinant protein | Nephronectin (human) | R&D Systems | 9560NP-050 | 1.5 µg/cm$^2$ |
| Peptide, Recombinant protein | Nephronectin (mouse) | R&D Systems | AF4298-NP-50 | 1.5 µg/cm$^2$ |
| Commercial assay or kit | CloneEZ PCR cloning kit | GenScript | Cat# L00339 | |
| Commercial assay or kit | DIG RNA Labelling Kit (SP6/T7) | Roche | Cat# 11175025910 | |
| Chemical compound, drug | PF-573228 | Sigma-Aldrich | Cat# PZ0117-5MG | 10 µM |
| Chemical compound, drug | Y27632 | Sigma-Aldrich | Cat# Y0503 | 10 µM |
| Other | DAPI stain | Roche | D8417 | (1 µg/mL) |
| Other | Phalloidin | Invitrogen | A-12380 | (1:200) |

## Chick embryos

All experiments were performed using fertilized White Leghorn chicken eggs (*Gallus gallus domesticus*) obtained from Texas A&M Poultry Center (College Station, TX). Eggs were incubated at 38°C in humidified conditions until the desired stages. Animal studies were approved by the Institutional Animal Care and Use Committee (IACUC) at Rice University.

For Npnt and Itga8 knockdown and overexpression studies, eggs were incubated for approximately 24–26 hr to obtain three-somite stage or HH8 (*Hamburger and Hamilton, 1951*), then windowed as previously described (*Spurlin and Lwigale, 2013*). A few drops of Ringer's solution containing 100 U/mL penicillin and 100 µg/mL streptomycin (PenStrep, Thermo Fisher Scientific) were added to embryos to maintain hydration. Embryos were injected with viral constructs (see below) using a Picospritzer III pneumatic microinjection system (Parker Hannifin) in the space between the vitelline membrane and cranial region, ensuring that the neural tube and adjacent ectoderm were completely covered (*Figure 2A*). Injected embryos were re-incubated and collected for GFP screening at the desired stages of development ranging between E5 and E15. Only the corneas showing robust GFP expression were used for subsequent analyses.

## Histology, H&E, and immunohistochemistry

Embryos were collected at desired stages and eyes were dissected in Ringer's saline solution. Samples collected for H&E staining were fixed overnight at 4°C in modified Carnoy's fixative (60% ethanol, 30% formaldehyde, and 10% glacial acetic acid). Afterward, the tissues were dehydrated in ethanol series, cleared in Histosol (National Diagnostics), embedded in paraffin blocks, and sectioned at 10 µm thickness. Sections were stained with hematoxylin for 15 s, counterstained with eosin for 40 s, dehydrated in ethanol series, and mounted with Cytoseal (Thermo Fisher Scientific) for imaging. For Npnt immunohistochemistry, eyes were fixed in methanol-acetic acid (MAA) fixative (98% methanol, 2% glacial acetic acid) that was chilled on dry ice. Samples were maintained in MAA fixative at –80°C for at least 2 days, then were gradually warmed to room temperature before

further dehydration in ethanol series and embedding in paraffin blocks. Samples were sectioned at 10 µm, rehydrated, then immunostained with Npnt antibody (1:100; orb221700, Biorbyt) following standard procedures. For all the other immunohistochemistry and phalloidin staining, samples were fixed in 4% paraformaldehyde (PFA) overnight at 4°C, embedded in paraffin, and sectioned as described above. Immunostaining with anti-GFP (1:500; A-6455, Invitrogen) and anti-fibronectin (1:30; B3/D6, Developmental Studies Hybridoma Bank) was used to detect protein expression. Explant cultures were immunostained with anti-phosphorylated paxillin (1:200; 44-722G, Invitrogen). Phalloidin-568 (1:200; A-12380, Invitrogen) and DAPI (Roche) were used to show total cell distribution and actin organization in whole-mount, sectioned tissue, and explant cultures. The following secondary antibodies (Invitrogen) were used at 1:200: Alexa-488 goat anti-rabbit IgG, Alexa-594 goat anti-rabbit IgG, Alexa-594 goat anti-mouse IgG2a, and Alexa-488 goat anti-mouse IgG1.

## Section in situ hybridization

Eyes were fixed in modified Carnoy's fixative, embedded in paraffin, and sectioned as described above. Riboprobes were generated from gene fragments using cDNA pooled from E7 anterior eyes and cloned into TOPO-PCRII (Invitrogen). Digoxigenin (DIG)-labeled riboprobes were synthesized following the manufacturer's protocol (DIG Labeling Kit, Roche). Primers used to generate the riboprobes are listed in *Supplementary file 1*. Sections were hybridized with riboprobe at 52°C (*Npnt*) and 60°C (*Itgα8*) overnight. Hybridization was detected using anti-DIG antibody conjugated with alkaline phosphatase (Roche) and color was developed with 5-bromo-4-chloro-3-indolyl phosphate/nitro blue tetrazolium (BCIP/NBT; Sigma). Following color development, sections were fixed with 4% PFA, mounted in Cytoseal, and imaged using an Axiocam mounted on an AxioImager2 microscope (Carl Zeiss).

## Production of RCAS virus

RCAS (*Hughes et al., 1987*) virus was used for stable and prolonged expression of shRNA constructs used for knockdown and the overexpression studies. To produce RCAS viral stocks, chick fibroblasts (DF-1 cells; lot 58217603 with no mycoplasma contamination detected, ATCC) were cultured in Dulbecco's modified Eagle medium (DMEM) supplemented with 10% fetal bovine serum (Invitrogen), 100 U/mL penicillin, and 100 µg/mL streptomycin (referred to here as complete DMEM, Thermo Fisher Scientific). Cells were transfected at about 70% confluency with plasmid DNA containing the RCAS constructs using Lipofectamine 3000 (Invitrogen). Transfected cells were grown for 4 days to enable viral replication. Media containing viral particles were collected on subsequent days, pooled, and centrifuged at 21,000 rpm (Beckman) for 1.5 hr at 4°C to concentrate the virus. Virus pellets were resuspended in DMEM and stocks at approximately $1–7 \times 10^6$ lfu/mL were stored in –80°C until use.

## Generation of shRNA and viral constructs

shRNA target sequences used for $Npnt^{kd}$ and $Itg\alpha8^{kd}$ (*Supplementary file 2*) were designed using the BLOCK-iT RNAi Designer tool (Thermo Fisher Scientific) and linked to their reverse complementary sequence by a loop sequence, TTCAAGAGA. A short termination sequence and restriction enzyme sites were added at each end to create fragments that were ligated into a pSLAX shuttle vector (*Hughes et al., 1987*) and to add a chick U6 (Cu6) promoter and a GFP reporter. The shRNA sequences were cloned into the RCAS vector (*Figure 2—figure supplement 1A and B*) by homologous recombination using a CloneEZ kit (GenScript) as previously described (*Kwiatkowski et al., 2017*; *Ojeda et al., 2017*). The $Npnt^{oe}$ overexpression constructs were either generated by substitution of the shRNA with a full-length coding sequence of Npnt in the vector above or using a modified vector where the full-length Npnt protein was driven by a promoter within the viral long terminal repeats (LTRs) with eGFP linked by an IRES motif (*Figure 2—figure supplement 1B*). Constructs containing either mutated Npnt with the RGD sequence changed to RAE (PR<u>GD</u>VFIPRQPGVSNNLFEIL<u>EI</u>ER to PR<u>AE</u>VFIPRQPGVSNNLFEIL<u>AIA</u>R) or the truncated version of Npnt containing only the N-terminal EGF-like repeats domain (Npnt-EGF) were commercially synthesized (GenScript) and cloned into the modified RCAS vector. All constructs were validated using primers for Npnt and Itga8 (*Supplementary file 3*).

## BrdU staining

BrdU was used to assay for cell proliferation. The BrdU solution was prepared at a final concentration of 10 μM in complete media. Eyes were collected at desired stages and injected into the anterior chamber between the cornea and lens with BrdU solution (approximately 50–100 μL depending on development stage), then cultured in BrdU solution for 2 hr at 37°C. Eyes were rinsed in phosphate-buffered saline , fixed in 4% PFA, embedded in paraffin, and sectioned as described above. BrdU-positive cells were identified by immunohistochemistry using anti-BrdU antibody (1:30; G3G4, Developmental Studies Hybridoma Bank). Sections were counterstained with DAPI.

## In vitro explant culture

Embryos were collected at E4 in Ringer's solution, and the anterior eyes were dissected and digested in dispase (1.5 mg/mL, Worthington) for 10 min at 37°C. The periocular mesenchyme was isolated by physically removing the presumptive cornea epithelium, lens, and the optic cup. The mesenchyme ring was further trimmed using tungsten needles to obtain cells that are proximal to the presumptive cornea, then dissected into approximately 120 × 120 μm explants. Nunc Lab-tek II 8-well chamber slides (Sigma) were coated with 1.5 μg/cm$^2$ with poly-D-lysine (MP Biomedicals) for 1 hr at room temperature, followed by recombinant mouse or human Npnt (R&D Systems) at 1.5 μg/cm$^2$ for 2 hr at 37°C. Mesenchyme explants were transferred to the coated slides containing complete media, with and without inhibitors and incubated at 37°C in a humidified tissue culture incubator with 5% $CO_2$ for 12–24 hr. The α8β1 peptide inhibitor following the 23-mer sequence P**RGD**VFIPRQPTN**DLFEIFEIER** (*Sato et al., 2009*) was commercially generated (GenScript) and used at a 10 μM working concentration. The small-molecule FAK inhibitor PF-573228 (Sigma; *Slack-Davis et al., 2007*) and ROCK inhibitor Y27632 (Sigma; *Uehata et al., 1997*) were used at 10 μM working concentration. Explants with migratory cells were imaged with a Rebel T6s camera (Canon) mounted on an Axiovert 40C microscope (Carl Zeiss). The magnitude of cell migration from explants was calculated by measuring the density of cells in a 175 μm × 175 μm area located 270 μm from the center of the explant. The orientation of pNC migration from explants was analyzed as previously described (*Babaliari et al., 2018*). Briefly, confluent regions of cells proximal to the explant were segmented and linearized, then the ImageJ Directionality plugin using local gradient orientation (*Liu, 1991*) was applied. The cellular features oriented to 90° ± 30° were considered to be aligned perpendicular to the explant. Mean integrated peak areas, bounded by 60° and 120°, were used to compare the magnitude of perpendicular alignment in all conditions.

## Time-lapse video microscopy

Mesenchyme explants were prepared and cultured as described above in media containing 0.1 μg/mL Hoechst dye solution (Thermo Fisher Scientific) used to label all nuclei. Chamber slides containing attached explants were imaged at 3 min and 27 s intervals for 17 hr using an FV1200 laser scanning microscope (Olympus) with a stage top incubator (Tokai Hit). Movies were generated using Imaris Software (Oxford Instruments).

## Quantification of corneal measurements

### Corneal thickness

Differences in corneal thickness between control and knockdown samples were determined by averaging measurements taken at three separate locations along the cornea. All measurements were taken perpendicular to the radial curvature of the cornea. Since corneal thickness is not always uniform in overexpression samples, measurements were taken at three locations across the center of the thickened region.

### Cornea cell counts and density

Nuclei labeled by the DAPI staining of corneal sections were used to count stromal cells within a width of 200 μm along the entire height of that region. Cells were either counted manually or using the threshold particle analysis in ImageJ software in which nuclei from the epithelium and endothelium are discounted from the final tally. Cell density was determined by dividing the number of nuclei and the area, which was measured by multiplying the value of the corneal thickness by the width of the selected region. A comparison of cell counts and density between the anterior vs. posterior corneal

region was determined by taking similar areas for all corneas, which was set at a value of 20% the average thickness of control corneas.

## Cell proliferation

Cell proliferation was determined in corneal sections by first utilizing the cell count technique described above. The total number of cells was represented by all the DAPI-positive nuclei, out of which the BrdU-positive cells were quantified. Cell proliferation was calculated as a percentage of the total DAPI-positive nuclei that were BrdU positive.

## Software and statistics

ImageJ software was used to measure the staining intensity or tissue morphological features, such as cornea thickness and cell density. The number of cornea sections analyzed is summarized in each figure legend. All statistical analyses were conducted using GraphPad Software. Data are presented as scatterplot with mean values. Statistical significance was determined by two-tailed unpaired Student's $t$-test and was used to compare the differences between means. Samples with p-values < 0.05 were considered significant.

## Acknowledgements

We thank the members of Lwigale lab for the helpful discussions and suggestions on this project. We also like to thank the Warmflash lab for use of the FV1200 laser scanning microscope for live imaging of periocular mesenchyme explants. This work was funded by the National Institutes of Health Grants R01 EY031381 and EY022158 (to PYL).

## Additional information

### Funding

| Funder | Grant reference number | Author |
|---|---|---|
| National Eye Institute | EY031381 | Peter Lwigale |
| National Eye Institute | EY022158 | Peter Lwigale |
| National Institutes of Health | National Institutes of Health | Peter Lwigale |

The funders had no role in study design, data collection and interpretation, or the decision to submit the work for publication.

### Author contributions

Justin Ma, Data curation, Formal analysis, Investigation, Methodology, Software, Validation, Writing - original draft, Writing - review and editing; Lian Bi, Data curation, Formal analysis, Investigation, Methodology; James Spurlin, Data curation, Formal analysis, Investigation, Methodology, Software, Writing - review and editing; Peter Lwigale, Conceptualization, Data curation, Formal analysis, Funding acquisition, Investigation, Methodology, Project administration, Supervision, Validation, Visualization, Writing - original draft, Writing - review and editing

### Author ORCIDs

Peter Lwigale ⬡ http://orcid.org/0000-0003-1799-4905

### Ethics

This study was conducted in strict accordance with the recommendations in the Guide for the Care and Use of Laboratory Animals of the National Institutes of Health. Fertilized chick embryos incubated between 1 to 17 days were handled according to the approved institutional animal care and use committee (IACUC) protocol (#IACUC-20-190) of Rice University.

### Decision letter and Author response

Decision letter https://doi.org/10.7554/eLife.74307.sa1

Author response https://doi.org/10.7554/eLife.74307.sa2

## Additional files

### Supplementary files
- Supplementary file 1. Table showing the primer sequences used for riboprobe synthesis.
- Supplementary file 2. Table showing the shRNA target sequences used for knockdown studies.
- Supplementary file 3. Table showing the primers used to validate *Npnt* and *Itgα8* knockdown efficiency.
- Transparent reporting form
- Source data 1. Statistical analysis reported in *Figures 2–7*.
- Source data 2. Gel images for the data reported in *Figure 2—figure supplement 1C*.

### Data availability
All data generated or analyzed during this study are included in the manuscript and supporting file; Source Data files have been provided for Figures 2-7.

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
