## [Editor Report]

This work investigates the role of extracellular matrix (ECM) component nephronectin (Npnt) and integrin a8 (Itga8) in the migration of periocular mesenchymal cells during vertebrate corneal development. They find that knockdown or overrexpression of Npnt and Itga8 leads to changes in corneal thickness, and their finding suggests that Npnt augments cell migration into the presumptive cornea ECM by functioning as a substrate for Itgα8-positive periocular neural crest.

---

## [Decision Letter]

**Decision letter after peer review:**

Thank you for submitting your article "Nephronectin-Integrin a8 signaling is required for proper migration of periocular neural crest cells during chick corneal development" for consideration by *eLife*. Your article has been reviewed by 2 peer reviewers, and the evaluation has been overseen by a Reviewing Editor and Marianne Bronner as the Senior Editor. The following individual involved in review of your submission has agreed to reveal their identity: Ales Cvekl (Reviewer #2).

In this manuscript the authors examine early stages in avian corneal development, focusing on how neural crest derived cells migrate into the space between the corneal epithelium and lens to form the corneal endothelium and stroma. The authors examine the functional interplay of nephronectin (Nptm) and alpha8-integrin (Itga8) and find that both overexpression and knockdown of Npnt and Itga8 Kd cause changes of corneal thickness, but do not significantly impact corneal transparency and function. In general the reviewers felt that this work would be suitable for *eLife* if some specific points could be addressed.

1) The authors should should examine the expression of corneal differentiation markers such as keratocan, lumican and collagen to determine if overexpression /knockdown of Npnt/ Itga8 impact corneal differentiation beyond cell migration.

2) The authors should provide data showing what signals/targets downstream of Npnt-integrin a8 underlie the attenuation of cell migration.

3) Please indicate if the corneas are transparent after over expression/ KD of Npnt /integrin 8a and discuss whether these manipulations lead to pathological phenotypes? (eg did they develop corneal disorder such as corneal ectasia, corneal opacity etc?)

4) There is some concern about the specificity/sensitivity of the Npnt antibody given that there is very strong signal in mRNA but little signal in protein level in several figures.For example there is abundant expression of mRNA but little Npnt protein detected in the corneal stroma from E7, E9 and E12. At E7, Npnt mRNA is highly expressed but Npnt protein is not detected. Similarly in Figure 5, the expression pattern of Npnt mRNA and protein are not consistent. Since Npnt is ECM component, it should be evenly distributed throughout the corneal stroma instead of punctate expression. Given these concerns, for the experiments in Figure 2, the authors should perform immunostaining to show that Npnt protein levels are indeed reduced.

5) In Figure 4B the first wave migration of pNC to form endothelium was attenuated by Itga8 KD but in Fig4D endothelium formation does not appear to have any defect. How do the authors explain this? One might have expected cornea-lens fusion due to the impairment of endothelium formation.

6) When talking about the range of 70-90 kDa Npnt proteins, it would be useful to state that these are generated by alternate splicing and give a range of aa residues (e.g. 536 to 595aa) to avoid confusion with complex posttranslational modifications.

7) At least four additional proteins interacting with NPNT include ESR2, FLT3, insulin and WDR76. Are these genes expressed in the system and what we can learn from their known roles in other systems for corneal cell biology?

---

## [Author Response]

In this manuscript the authors examine early stages in avian corneal development, focusing on how neural crest derived cells migrate into the space between the corneal epithelium and lens to form the corneal endothelium and stroma. The authors examine the functional interplay of nephronectin (Nptm) and alpha8-integrin (Itga8) and find that both overexpression and knockdown of Npnt and Itga8 Kd cause changes of corneal thickness, but do not significantly impact corneal transparency and function. In general the reviewers felt that this work would be suitable for eLife if some specific points could be addressed.1) The authors should should examine the expression of corneal differentiation markers such as keratocan, lumican and collagen to determine if overexpression /knockdown of Npnt/ Itga8 impact corneal differentiation beyond cell migration.

The reviewers raise an important point, which we are currently studying at later stages of corneal development. To avoid an excessively large manuscript, the current study only focused on the role of Npnt during periocular neural crest migration. To address the reviewers concern, we would like to share some of our preliminary data (see Author response image 1) showing that keratocyte differentiation is not disrupted at E8 following overexpression of *Npnt*. The keratocyte markers Coll2, KSPG, and Decorin are all expressed although they appear at relatively less intense levels following overexpression of Npnt. We also examined the basement membrane proteins Perlecan and Laminin, which were strongly expressed in the epithelial basement membrane, but appeared to be affected in the Descemet membrane. We addressed the reviewers comment by adding the statements below in the Discussion section of the revised manuscript.

**Author response image 1. sa2fig1:** 

Page 17; Line 384-386:

“However, additional studies may be required to determine how the disruption in corneal thickness observed in the current study may affect the corneal ECM and transparency at later stages of development.”

Page 19; Lines 426-436:

“In this study, we focused on the role of Npnt/Itga8 signaling during pNC migration. Our observation that the Npnt protein subsequently localizes to the basement membrane of the epithelial layer combined with its pleiotropic functions during development and in disease (Yamada and Kamijo, 2016), imply that Npnt may have other functions at later stages of cornea development. One possibility is that Npnt mediates epithelial-mesenchymal interactions between the cornea epithelium and stroma by signaling through the EGF-like domain to EGFR receptors in the epithelial cells. Future studies will expand on these findings to determine whether the corneal thinning and thickening phenotypes observed in the current study affect transparency, which develops at later stages. Elucidation of the link between Npnt signaling and corneal cell differentiation may provide useful insights for future therapeutic applications for wound healing and regeneration studies.”

2) The authors should provide data showing what signals/targets downstream of Npnt-integrin a8 underlie the attenuation of cell migration.

We thank the reviewers for the excellent suggestion that substantially improved the paper. To investigate the downstream signals, we conducted pNC explant culture experiments on Npnt substrate and focused on focal adhesion kinase (FAK inhibitor), focal adhesion complex (phospho paxillin), and Rho-associated kinase (ROCK inhibitor), and generated a new Figure 7 and Figure 7—figure supplement 1. Our results indicate that Npnt/Itga8 signaling induces pNC migration through activation of Rho mediated by FAK and paxillin. We addressed the new findings in the Results section (Pages 13-14; Lines 277-312), Discussion section (Pages 18-19; Lines 404-416), and also in various parts of the Methods section.

3) Please indicate if the corneas are transparent after over expression/ KD of Npnt /integrin 8a and discuss whether these manipulations lead to pathological phenotypes? (eg did they develop corneal disorder such as corneal ectasia, corneal opacity etc?)

We thank the reviewers for this comment. In the current study, we focused on the early events of cornea development and analyzed the corneal defects prior to development of corneal transparency, which occurs gradually between E15-19 in the chick. As part of our ongoing studies, we plan to analyze whether corneal thinning that follows knockdown of *Npnt* or *Itga8* persists into later stages of development and if so, we will determine whether this defect leads to corneal ectasia and/or loss of transparency. We will also examine whether the corneal thickening defect affects transparency. To address the reviewers comment, we are sharing some of our promising preliminary data from E15 overexpression studies showing substantial loss of transparency, defects in the iris (Author response image 2, arrow) indicated by the large pupil and coloboma (arrowhead), and in some cases, apparent increase in corneal diameter. We addressed this concern in the revised manuscript in our response to comment 1 above.

4) There is some concern about the specificity/sensitivity of the Npnt antibody given that there is very strong signal in mRNA but little signal in protein level in several figures.For example there is abundant expression of mRNA but little Npnt protein detected in the corneal stroma from E7, E9 and E12. At E7, Npnt mRNA is highly expressed but Npnt protein is not detected. Similarly in Figure 5, the expression pattern of Npnt mRNA and protein are not consistent. Since Npnt is ECM component, it should be evenly distributed throughout the corneal stroma instead of punctate expression. Given these concerns, for the experiments in Figure 2, the authors should perform immunostaining to show that Npnt protein levels are indeed reduced.

We thank the reviewers for this comment and agree that there is a difference in the distribution of the mRNA versus the protein. In fact, one of the patterns we see is that despite strong mRNA expression in the stroma, the majority of the protein accumulates in the anterior region and subsequently localizes to the basement membrane of the epithelial layer. This is not surprising since Npnt localizes to the basement membranes during development of several tissues including the kidney, teeth, and skin. To address this part of the reviewers’ concern, we added the clarification below elaborating on the localization of the protein vs mRNA expression in the Results section (Page 7; Lines 142-150).

“Interestingly, the protein expression did not correlate with the strong mRNA expression in the stroma during later stages of cornea development. This mis-match could be due to posttranscriptional regulation that prevents protein expression. It is also possible that posttranslational modification by enzymes such as matrix metalloproteinases (MMPs), which are temporally and spatially regulated in the corneal ECM during development (Huh et al., 2007), could lead to protein degradation. A previous study showed that Npnt can be modified by MMP cleavage (Toraskar et al., 2019). Given that the changes in protein localization occur after the second wave of migration, we can conclude from our results that expression of Npnt coincides with pNC ingression into the cornea, implicating a potential role during development.”

To address the second part of the reviewers’ concern, we would also like to refer to the dynamic distribution of the Npnt that localizes the protein in the epithelial basement membrane at E7 and later stages of development, which makes it difficult to distinguish Npnt protein levels in the stroma of control vs knockdown. For this reason, we chose to use mRNA expression to assess the change in *Npnt* expression induced by our knockdown construct.

5) In Figure 4B the first wave migration of pNC to form endothelium was attenuated by Itga8 KD but in Fig4D endothelium formation does not appear to have any defect. How do the authors explain this? One might have expected cornea-lens fusion due to the impairment of endothelium formation.

We agree with the reviewers’ concern that our experiments do not show defects related to absence of the corneal endothelium at E5. A potential reason is that the pNC contain a heterogenous population that either express or do not express *Itga8* (Figure 3A), and in the *Itga8* knockdown experiments, some of the cells that do not endogenously express Itga8 may compensate for this loss of cell migration and form a normal endothelial cell layer. A similar explanation applies to the second wave, although in this case, the corneal thinning defect persists due to the large number of cells required for the stroma. We clarified this in the Results section of the manuscript as indicated below.

Page 9; Lines 201-205:

“Despite the attenuated migration of pNC expressing the Itga8^kd^ construct during the first wave, we did not observe defects in the corneal endothelium. One possibility is that pNC which do not endogenously express Itga8 (Figure 3A), may also contribute to the corneal endothelium albeit at a lower level, but they are able to compensate for the Itga8 knockdown resulting in the formation of a normal endothelial layer.”

Page 10; Lines 211-214:

“As indicated in our analysis at E5, it is likely that non-Itga8 expressing pNC may compensate during the second wave of pNC migration, but not to the extent that abrogates the corneal thinning defect, possibly due to the relatively large number of cells required for the formation of the stroma.”

6) When talking about the range of 70-90 kDa Npnt proteins, it would be useful to state that these are generated by alternate splicing and give a range of aa residues (e.g. 536 to 595aa) to avoid confusion with complex posttranslational modifications.

We thank the reviewers for the comment. We agree with the reviewers that the ranges in sizes of the Npnt proteins is generated by alternate splicing. We cited the sizes of Npnt from one of the initial studies that discovered this protein in the mouse (Brandenberger et al., 2001). Interestingly this study only provided two isoforms (561, 578 aa), although currently there are four isoforms of mouse Npnt (561, 578, 592, and 609 aa). To accurately cite the study and provide the range of amino acids, we edited the sentence as indicated below.

Page 4: lines 82-85:

“Npnt was discovered as an ECM ligand for integrin a8b1 (a8b1) during mouse kidney development, consisting of 70-90 kDa proteins (Brandenberger et al., 2001) generated by alternate splicing (561 to 609 amino acids).”

7) At least four additional proteins interacting with NPNT include ESR2, FLT3, insulin and WDR76. Are these genes expressed in the system and what we can learn from their known roles in other systems for corneal cell biology?

We thank the reviewers for their insightful comment that brought attention to the recent publication of the human interactom (Huttlin et al., 2021), which identified 10 proteins that interact with Npnt including ITGA8, *NOTCH2*, CRLF1, ESR2, IGFL1, INS, LY6G5C, TAL1, WDR76, and FLT3. We agree with the reviewers that it would be interesting to explore other proteins that may interact with Npnt in other contexts of corneal development, but in the current study we focused on Npnt/Itga8 and the role of this interaction on pNC migration during early development of the cornea. Nonetheless, we identified from literature searches that among these 10 proteins, *NOTCH2*, ESR2, IGFL1 and INS are expressed in the cornea epithelium.

Furthermore, our previous RNAseq data (Bi and Lwigale, 2020) indicate that ESR2 and WDR76 are expressed by the pNC and during their differentiation into corneal endothelium and keratocytes. Whereas INS transcripts are upregulated only in the corneal endothelium. We are grateful for this comment and are looking forward to incorporate these proteins in future studies addressing the potential role of Npnt on pNC differentiation and later stages of corneal development.